∂ | **Open Peer Review** | Bacteriology | Research Article

# Optimized use of the FilmArray Meningitis/Encephalitis panel for early discontinuation of antibiotic therapy

Nathan Nicolau-Guillaumet,[1] Marin Moutel,[2] Chloé Plouzeau-Jayle,[3] Gauthier Pean de Ponfilly,[4,5] VIrginie Courbin,[6] Anne-Gaëlle Ranc,[7] Hélène Revillet,[8] Bruno Mourvillier,[9] Maxime Hentzien,[2] Anaëlle Muggeo,[1] Thomas Guillard,[1] On behalf of the GMC Study Group

**ABSTRACT** Suspected meningitis/encephalitis cases require prompt empirical antimicrobial therapy. Syndromic diagnostics like the FilmArray Meningitis/Encephalitis (FAME) panel offer rapid results and are increasingly used despite high costs. This study assessed FAME's integration in clinical microbiology laboratories' everyday practice and its impact on early treatment decisions. The FilmArray Meningitis/encephalitis panel Outcome Study (FAMOuS) retrospectively reviewed biological data, treatments before/after FAME, and outcomes from 783 patients across six French hospitals over a 3-year period. FAME results were 80% negative, 7% bacterial, and 13% viral, with no bacterial false negatives. Positive results in samples with <10 leukocytes were exclusively seen in immunocompromised individuals and children under 2. Positive results prompted therapeutic changes in 74% of cases, compared to 50% for negatives. FAMOuS enabled the proposal of a decision-support flowchart to optimize the rational use of FAME and support antibiotic stewardship, with the potential to increase early antibiotic discontinuation and reduce unnecessary testing.

**IMPORTANCE** Prompt identification of infectious causes is essential in suspected meningitis and encephalitis, where timely treatment decisions can be lifesaving. The FAMOuS study evaluated the use of the FilmArray Meningitis/Encephalitis (FAME) panel across six French hospitals, analyzing nearly 800 patient episodes. The findings demonstrate that FAME offers reliable results that directly influence clinical management—guiding therapeutic adjustments in most positive cases and supporting early antibiotic discontinuation when appropriate. The study highlights specific patient contexts where FAME provides the greatest clinical value and proposes a decision-support flowchart to optimize its rational use. By integrating rapid molecular diagnostics into everyday practice, this work contributes to improved patient care and reinforces hospital antibiotic stewardship strategies.

**KEYWORDS** FilmArray, meningitis, encephalitis, antibiotic stewardship , decision-making flowchart

Meningitis and encephalitis (ME) remain major global health problems with significant morbidity and mortality and constitute medical emergencies in which rapid etiological diagnosis and early antimicrobial treatment are essential (1, 2). Conventional microbiological methods are specific but often lack sensitivity after prior antibiotic exposure and require prolonged processing times, which force clinicians to initiate or maintain empirical broad-spectrum therapy while awaiting results. Median times to pathogen identification with standard techniques have been reported near 96 h, a critical window during which unnecessary antimicrobial exposure can increase adverse events and the risk of colonization by resistant organisms (3–5). Syndromic

Address correspondence to Thomas Guillard, tguillard@chu-reims.fr.

The authors declare no conflict of interest.

molecular testing, such as the FilmArray Meningitis/Encephalitis (FAME) panel (bioMérieux, France), performs multiplex PCR for 14 pathogens on cerebrospinal fluid (CSF) and reduces median time to identification to approximately 2 h (3, 6). Despite high reported sensitivity and specificity for common targets, the FAME panel raises practical issues related to cost, incomplete organism coverage, and the need to interpret results in a clinical context (6, 7). Published work shows mixed effects on clinical outcomes: reductions in hospital length of stay in adults and decreases in antimicrobial duration mainly in pediatric cohorts (8). Overall, randomized trials evaluating the FAME and, hence, high-quality evidence are lacking. Evidence is still confined to observational, usually retrospective, and single-center studies (9). Current guidelines therefore recommend targeted PCRs in specific situations rather than routine syndromic testing (10–14). The aims of our retrospective multicenter study were to establish evidence-based criteria for targeted FAME use and to evaluate the panel's early impact on therapeutic decision-making, with particular emphasis on antimicrobial stewardship.

## MATERIALS AND METHODS

### Study design

Patients of any age from six French tertiary care hospitals (Lyon, Paris Bégin, Paris Saint Joseph, Poitier, Reims, and Toulouse), who underwent FAME according to the manufacturer's instructions on a lumbar puncture between January 2019 and December 2022, were retrospectively included in the FAMOuS study (FilmArray Meningitis/Encephalitis panel Outcome Study). The following clinical characteristics were collected: gender, age, hospitalization unit, immunodepression status (Table S1), time between admission and FAME, length of hospital stay, CSF biochemistry, CSF microbiology (cell counts, direct examination, and culture), FAME result, dexamethasone treatment before and after FAME, hospital-acquired or community-acquired status, specific viral PCR result if performed, death within 30 days, meningeal-targeted antibiotic and antiviral treatments administered prior to FAME and within the first 24 h after receiving the FAME results, as well as the duration of these treatments. Suspicion of hospital-acquired ME was defined as the onset of symptoms occurring more than 48 h after a patient was hospitalized. Based on the data "treatments before/after FAME," therapeutic de-escalation was defined as discontinuation of treatment, a reduction in the number of antibiotics used, or a narrowing of the spectrum of β-lactams according to the classification of Weiss et al. (15). Conversely, therapeutic escalation was characterized by an increase in the number of antibiotics or a broadening of the β-lactam spectrum according to Weiss et al. (15). No impact was considered when the treatment remained unchanged. In cases of positive direct examination, the impact of the FAME result was deemed non-assessable, as the direct examination already guided the therapeutic approach. Positive PCR or culture results were considered contaminations when supported by a set of arguments, including CSF biochemistry not consistent with infection and, more importantly, a clinical decision ruling out infection with a favorable outcome without treatment or with an alternative diagnosis. In cases of incomplete patient data, they were still included in the analyses for which the available data were sufficient. The collected data were harmonized for analysis (see Supplementary Material). Criteria for FAME performing in each center were also recorded.

### Statistical analysis

The data are expressed as mean values ± standard deviation, median (min − max), or numbers (percentages) when appropriate. Comparisons were performed using the $\chi^2$ test or Fisher's exact test for qualitative variables and the $t$-test or Mann–Whitney test for quantitative variables when appropriate. Univariate and multivariate analyses using conditional logistic regression were conducted to identify independent risk factors for positive FAME. Univariate analysis was performed to select candidate variables for

inclusion in the multivariable analysis. Variables with a *P*-value < 0.20 in the univariate comparison between FAME-positive and FAME-negative cases were included in the multivariate model. A *P*-value < 0.05 was considered significant.

## RESULTS

### Population and FAME results

From six tertiary centers, 786 patients underwent FAME testing performed 24/7 at clinicians' requests (details are provided in Table S2). Most ME suspicions predominantly came from adult emergency departments, pediatric emergency departments, intensive care, and neurology units (Fig. 1B). Three tests were invalid and excluded. Overall, 80% of FAME assays were negative, 7% were positive for bacteria (*n* = 55), 13% were positive for viruses (*n* = 100), and one sample was positive for *Cryptococcus*. Patients' characteristics are detailed in Table 1. Patients were predominantly male (55%) with a median age of 38 years; 14% were immunocompromised. The median delay from admission to FAME testing was 1 day. Average length of stay was 16.5 days (±36).

CSF white blood cell (WBC) count and biochemical parameters, including glucose, protein, and particularly lactate, were significantly associated with bacterial FAME positivity and with FAME positivity overall (see Fig. S1 and Table 1). Administration of antibiotics or dexamethasone prior to FAME was also significantly associated with a positive FAME result, particularly for bacteria (Table 1). In subgroup analyses, none of the suspected hospital-acquired ME cases yielded a positive FAME; similarly, samples from neurosurgery were uniformly negative in our series (Fig. 1A).

### Diagnostic markers and thresholds

Bacteria-positive FAME cases exhibited markedly higher CSF WBC counts than virus-positive cases and negative cases (median = 1,530 [2–21,000] vs 135 [2–2,300] and 17 [1–12,700], *P* < 0.0001). Given age-dependent variations in CSF cell counts, we analyzed WBC and FAME positivity by age and observed no true bacterial positives at low cell counts across age strata and a reduced likelihood of FAME positivity beyond age 70 (see Fig. S2). When stratifying by WBC thresholds, under 20 WBC/mm$^3$, there were 316 negatives, 16 viral, and 3 bacterial positives; under 10 WBC/mm$^3$, there were 253 negatives, 10 viral, and the same 3 bacterial positives (see Fig. S3 and Table S3). Two of these three bacterial detections were subsequently classified as contaminations with *Haemophilus influenzae* according to biochemistry results and clinical decision, and one represented *Listeria monocytogenes* in an immunocompromised patient with concordant direct examination. As shown in Fig. S1 and Table 1, lactatorrhachia was markedly higher in bacteria-positive FAME, with a median of 11.6 mmol/L, compared with 2.3 mmol/L and 2.4 mmol/L for viral and negative cases (*P* < 0.0001). Traumatic lumbar puncture, as judged by RBC count, did not reliably exclude positive FAME results (see Fig. S3E).

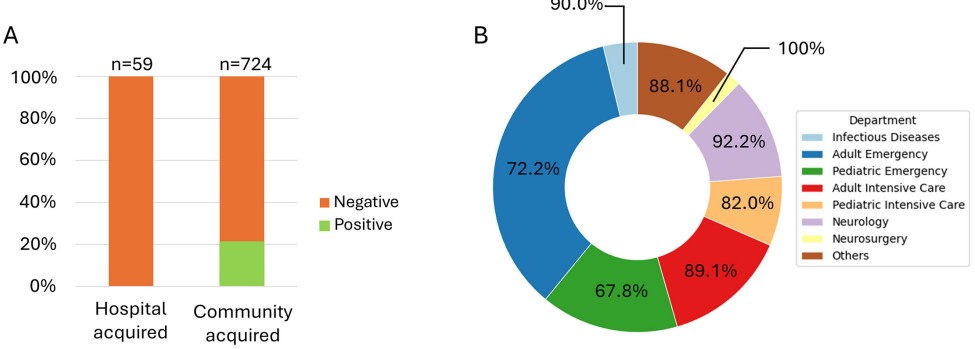

**FIG 1** Overview of FAME results across departments and acquisition settings. (A) Distribution according to acquisition type. (B) Distribution of FAME according to departments. The percentages of negative results are indicated for each department.

**TABLE 1** Characteristics of the 783 patients with interpretable FAME results included in the FAMOuS study[a]

| | Total | FAME positive | | | | FAME negative | P value[c] |
|---|---|---|---|---|---|---|---|
| | | Bacteria | Virus | P value[b] | Yeast | | |
| n | 783 | 55 | 100 | | 1 | 627 | |
| Age, years | 38 (0–97) | 46 (0–88) | 30 (0–94) | 0.175 | 88 | 41 (0–97) | <0.0001 |
| Age <2 years old | 134 | 15 | 21 | 0.376 | 0 | 98 | <0.05 |
| Men | 431 | 35 | 51 | 0.130 | 1 | 344 | 0.839 |
| Immunosuppression | 107 | 13 | 7 | <0.005 | 1 | 86 | 0.770 |
| Admission time to FAME testing delay, days | 1 (0–215) | 0 (0–9) | 0 (0–36) | 0.461 | 14 | 0 (0–215) | <0.0001 |
| Hospital-acquired suspected cases | 59 | 0 | 0 | | 0 | 59 | <0.0001 |
| Length of hospital stay, days | 16.5 ± 36 | 18 ± 21 | 15.5 ± 74.5 | 0.750 | 35 | 16.5 ± 27 | 0.978 |
| CSF glucose (mmol/L) | 3.15 (0–3,009) | 0.33 (0–6.8) | 2.96 (0.35–121) | <0.0001 | 0.32 | 3.27 (0–3,009) | <0.0001 |
| CSF protein (mg/L) | 540 (90–20,470) | 900 (131–17,070) | 600 (131–7,785) | <0.05 | 3,500 | 520 (90–20,470) | <0.005 |
| CSF lactate (mmol/L) | 2.5 (1–26.3) | 11.6 (2.3–26.3) | 2.3 (1.1–5.2) | <0.0001 | 5.5 | 2.4 (1–22.9) | <0.0001 |
| CSF lactate lacking data (%) | 367 (47%) | 18 (33%) | 56 (56%) | | 0 | 293 (47%) | |
| CSF WBC (/mm$^3$) | 28 (2–21,000) | 1,530 (2–21,000) | 135 (2–2,300) | <0.0001 | 342 | 17 (2–12,700) | <0.0001 |
| WBC < 10/mm$^3$ | 266 | 3 | 10 | 0.329 | 0 | 253 | <0.0001 |
| CSF RBC (/mm$^3$) | 100 (100–5,000,000) | 291.5 (100–27,250) | 100 (100–290,000) | | 100 | 100 (100–5,000,000) | |
| Negative CSF Gram stain | 747 | 30 | | | 1 | 616 | |
| Positive CSF Gram stain | 36 | 25 | 0 | <0.0001 | 0 | 11 | <0.0001 |
| Consistent with FAME | 25 | 25 | | | | 0 | |
| Not consistent with FAME | 11 | 0 | | | | 11 | |
| Negative culture | 743 | 28 | 100 | <0.0001 | 1 | 615 | <0.0001 |
| Positive culture | 40 | 27 | | <0.0001 | | 12 | <0.0001 |
| In FAME panel | 29 | 27 | | | | 1 | |
| Off FAME panel | 11 | 0 | | | | 11 | |
| Patient with antibiotic before FAME | 393 | 52 | 44 | <0.0001 | 0 | 297 | <0.005 |
| Antibiotics duration, days | 4 (0–61) | 10 (0–34) | 1 (0–25) | <0.0001 | | 4 (0–61) | 0.057 |
| Patient with antiviral before FAME | 268 | 22 | 39 | 0.903 | 0 | 207 | 0.151 |
| Antivirals duration, days | 2 (0–48) | 0 (0–1) | 8 (8–22) | <0.005 | | 2 (0–48) | 0.672 |
| Patient with dexamethasone before FAME | 93 | 28 | 7 | <0.0001 | 0 | 58 | <0.0001 |
| Adding dexamethasone after FAME result | 6 | 3 | 0 | | 0 | 3 | |
| Stopping dexamethasone after FAME result | 45 | 8 | 7 | | 0 | 30 | |
| Antibiotic de-escalation in the 24 h following FAME results | 218 | 28 | 36 | 0.071 | 0 | 154 | <0.0001 |
| Antibiotic escalation in the 24 h following FAME results | 32 | 6 | 0 | <0.005 | 0 | 26 | 0.865 |
| Antiviral de-escalation in the 24 h following FAME results | 124 | 22 | 14 | <0.0001 | 0 | 88 | <0.05 |
| Antiviral escalation in the 24 h following FAME results | 12 | 0 | 5 | 0.092 | 0 | 7 | 0.057 |
| Death within 30 days | 44 | 7 | 0 | <0.0001 | 0 | 37 | 0.492 |

[a]Values are n, mean ± SD and median (min − max). CSF, cerebrospinal fluid; WBC, white blood cell; RBC, red blood cell.
[b]Comparison between bacteria- and virus-positive FAME.
[c]Comparison between positive and negative FAME.

In multivariate analysis, independent correlates of FAME positivity included lower likelihood for hospital-acquired ME (OR: 0.05, CI: 0.05–0.67), higher CSF lactate (OR: 1.24, CI: 1.04–1.47), higher CSF glucose (OR: 1.03, CI: 1.00–1.05), younger age (OR: 0.98, CI: 0.96–0.99), sex (OR: 0.35, CI: 0.18–0.66), and CSF WBC <10/mm$^3$ being protective (OR:

0.06, CI: 0.01–0.26; Fig. 2). ROC analysis identified an optimal CSF lactate cut-off of 6.6 mmol/L for predicting bacterial infection (see Fig. S4), which confirms the safe use of 3.2 mmol/L (14).

## FAME reliability and discordances

Among the 783 evaluable samples, one negative FAME corresponded to an in-panel culture positive for *H. influenzae* considered laboratory contamination. Eleven cultures grew off-panel organisms, including a non-K1 *Escherichia coli*; 6 were contaminated and 5 were considered true meningitis, 3 of which were hospital acquired. One case exhibited a negative FAME with positive HSV-1 PCR. On the other side, 10 cases considered true meningitis were positive in FAME and negative in culture: 3 *E. coli* K1, 3 *H. influenzae*, and 4 *Streptococcus agalactiae*. Data are summarized in Table S4.

## Impact on antimicrobial therapy

We evaluated therapeutic changes occurring within 24 h of FAME result availability, before culture results, to attribute modifications to the FAME. Among patients with negative direct examination, a negative FAME led to escalation in 4% ($n = 26$) of cases, de-escalation in 25% ($n = 151$), and no impact in 71% ($n = 439$). Bacterial FAME positivity prompted escalation in 7% ($n = 2$), de-escalation in 60% ($n = 18$), and no change in 33% ($n = 10$), whereas viral FAME positivity produced no escalation, de-escalation in 36% ($n = 36$), and no impact in 64% ($n = 64$; Fig. 3A).

Excluding patients who received no treatment before or after FAME (assuming a low clinical suspicion of meningitis by the physicians) increased apparent de-escalation: negative FAME de-escalation rose to 50%, bacterial FAME to 62%, and viral FAME to 82% (Fig. 3B). Overall, positive FAME results (viral or bacterial) led to de-escalation in 74% of cases, whereas negative FAME results led to de-escalation in about 50% of treated cases. When de-escalation occurred, it most frequently involved complete discontinuation of antibiotics rather than a simple narrowing of the therapeutic spectrum (Fig. 3C). Among patients with negative FAME who remained on antibiotics ($n = 132$), available lactate measurements ($n = 77$) showed that the majority had values below 6.6 mmol/L ($n = 66$) and many below 3.2 mmol/L ($n = 43$). Similarly, among the seven patients with a FAME positive for a virus in whom antibiotic therapy was not discontinued, lactate levels were measured in four of them—all of whom had concentrations below 3.2 mmol/L.

## Impact on antiviral therapy

Of the 268 patients receiving antiviral treatment prior to the FAME analysis, treatment was discontinued within 24 h post-FAME in 124 cases. Specifically, it occurred in 88 cases (70%) following a negative FAME result, in 14 cases (11%) after a positive FAME result for a virus (10 Enterovirus, 1 VZV, and 3 HSV-2), and in 22 cases (18%) after a FAME-positive result for a bacterium. Notably, no antiviral treatment was continued after 24 h with a positive FAME for a bacterium.

## DISCUSSION

Syndromic panels like FAME offer dramatically faster results and high performance for included targets but present dilemmas of cost, incomplete coverage, and interpretative challenges, notably with positive HHV-6 in non-immunocompromised patients and *H. influenzae* false positive (16–18). Current guideline recommendations for targeted PCR after negative direct examination may delay diagnosis when specific PCRs are not continuously available. In our cohort, such an approach would have missed 10 (18%) bacterial meningitis cases caused by *H. influenzae*, *S. agalactiae*, and *E. coli* K1 (10–14). Conversely, indiscriminate use of FAME is costly and sometimes non-contributive. As laboratories cannot sustain multiple molecular techniques for cost reasons, syndromic PCR panels, which allow random-access testing and are well suited to emergency diagnostics, represent a practical alternative when used judiciously and in appropriate clinical contexts. The FAMOuS study included 783 patients who underwent FAME, and

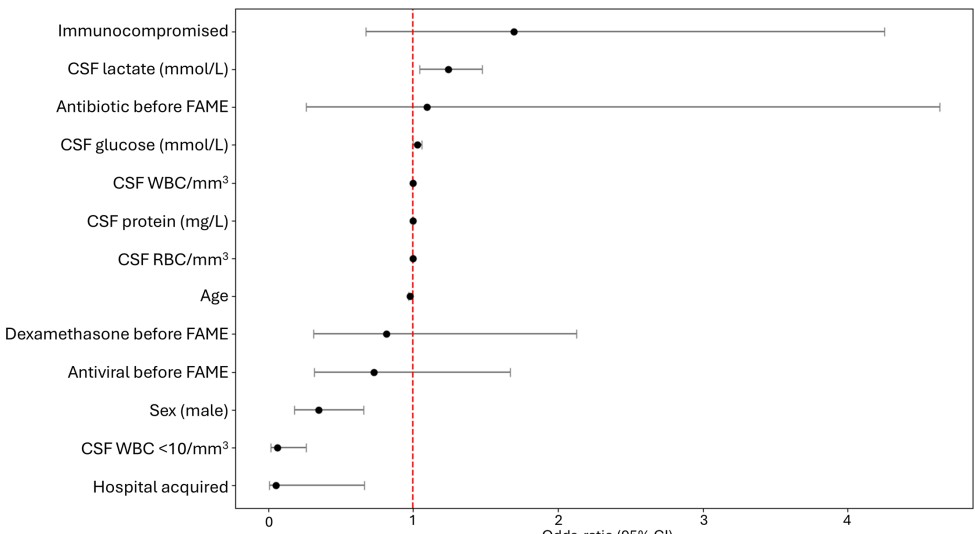

**FIG 2** Forest plots of main factors associated with FAME positive results. CSF, cerebrospinal fluid; FAME, FilmArray Meningitis/Encephalitis panel; RBC, red blood cells; WBC, white blood cells.

we reviewed biological data, treatments before/after FAME, and outcomes to assess integration in clinical microbiology laboratories and impact on early treatment decisions. Clinical data, such as headaches, photophobia, phonophobia, fever, or neck stiffness, were not collected for several reasons: (i) the clinical presentation can vary significantly from one patient to another, can be difficult to assess objectively, and may even be absent in some cases (19), (ii) there was a high risk of bias due to the retrospective nature of data collection, and (iii) gathering this information was beyond the scope of our study, as we focused on obtaining FAME results after the lumbar puncture, following the clinical suspicion of ME. By combining our data with literature thresholds, we propose a pragmatic, cost-conscious decision tree to concentrate FAME testing where it most increases diagnostic yield and where its rapid results can safely drive early discontinuation of unnecessary antibiotics (Fig. 4).

We recommend reserving first-line FAME use to community-acquired ME and performing the panel systematically in immunocompromised patients and in children under 2 years, regardless of CSF WBC, because pleocytosis may be absent in these groups (20–22). For other patients, a CSF WBC threshold of ≥10/mm$^3$, as proposed in several studies, identifies those with a higher likelihood of panel-detectable infection (23, 24); WBC <10/mm$^3$ generally yields no bacterial detection and should not prompt routine FAME testing for bacterial meningitis, especially with no biochemistry abnormalities (25). Given FAME's limited sensitivity for HSV, the absence of pleocytosis in some viral infections, and the possibility of early false-negative HSV PCR results, it remains essential to perform specific viral PCRs, notably repeating testing around day 4 after symptom onset (7, 26–28).

Traumatic lumbar puncture can spuriously raise CSF WBC, so the white cell-to-red cell ratio may help interpret cytology and guide the use of FAME testing (29); nevertheless, we observed FAME positives even with very high RBC counts. The data did not allow for determining whether FAME detected organisms originating from blood, but concordant positive FAME and CSF cultures in some cases support true meningitis rather than isolated bacteremia. CSF lactate provides an independent and strong discriminator: values above 6.6 mmol/L indicate a high probability of bacterial infection and warrant continuation or adaptation of empiric antibiotics, whereas values below 3.2 mmol/L support early antibiotic discontinuation when FAME is negative (30, 31).

As the literature struggles to demonstrate a clear impact of the FAME panel on the duration of antibiotic therapy, we initially sought to evaluate this effect within our own

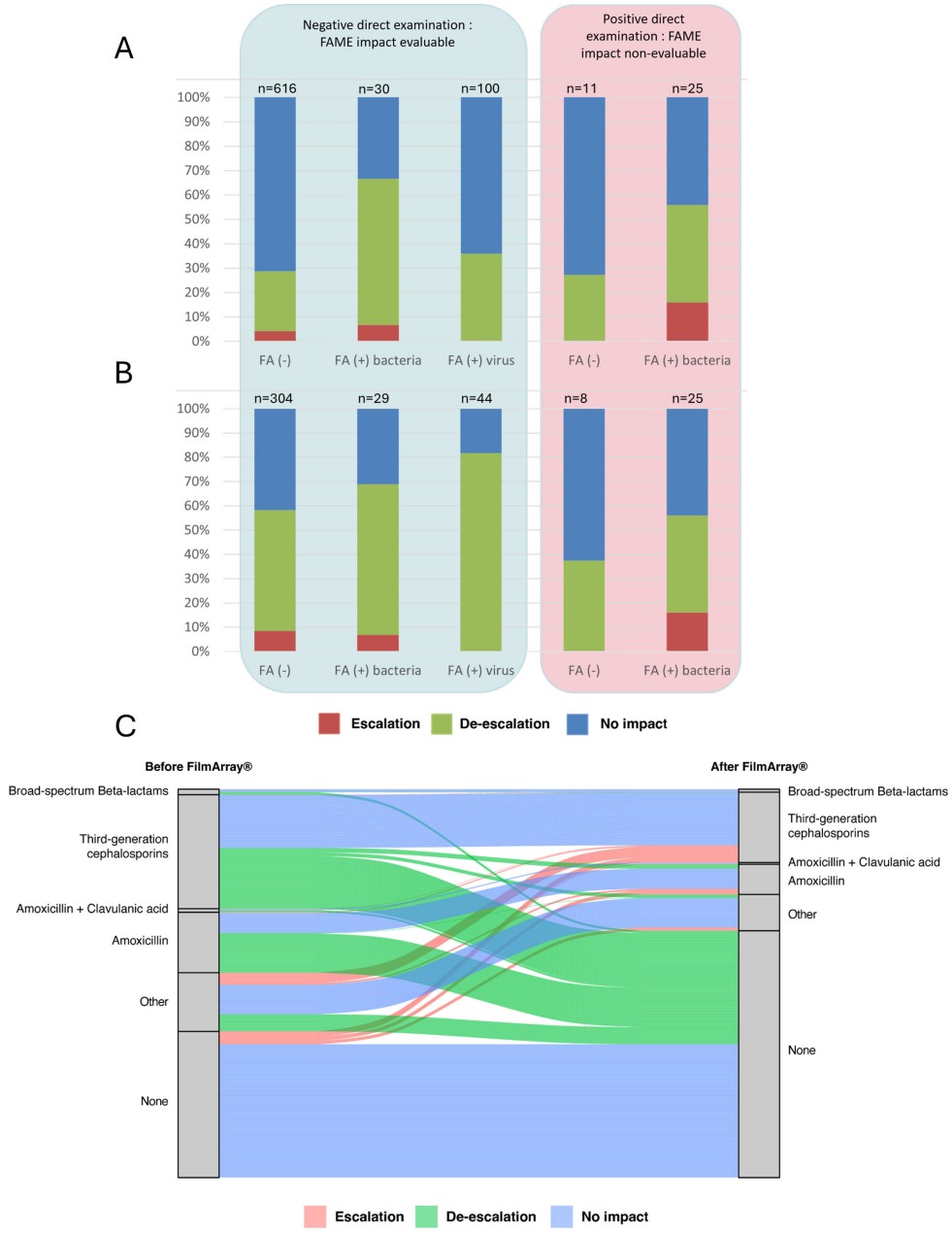

**FIG 3** FAME result impact on antibiotic therapy. (A) All patients. (B) Patients with antibiotic therapy administered either before or after the FAME results. (C) Sankey diagram of antibiotic treatment evolution between before and after FAME.

data set. FAME enabled a de-escalation of antibiotic therapy in many cases, especially when it was positive for a bacterium or a virus. Conversely, we found little impact on antibiotic therapy when the FAME result was negative. Indeed, a significant proportion (50%) of patients maintained their antibiotic therapy even though there was no false negative with the FAME.

Regarding patients on antiviral treatment, treatment was stopped in 46% of cases following the FAME, with 70% of these discontinuations occurring after a negative FAME result. Due to the necessity of a control viral PCR, evaluating the FAME's impact on antiviral therapy was challenging. Nonetheless, when the FAME was positive for an organism that did not require antiviral treatment, antiviral therapy was stopped in 100% of cases—demonstrating that the FAME may facilitate the early termination of unnecessary antiviral treatment.

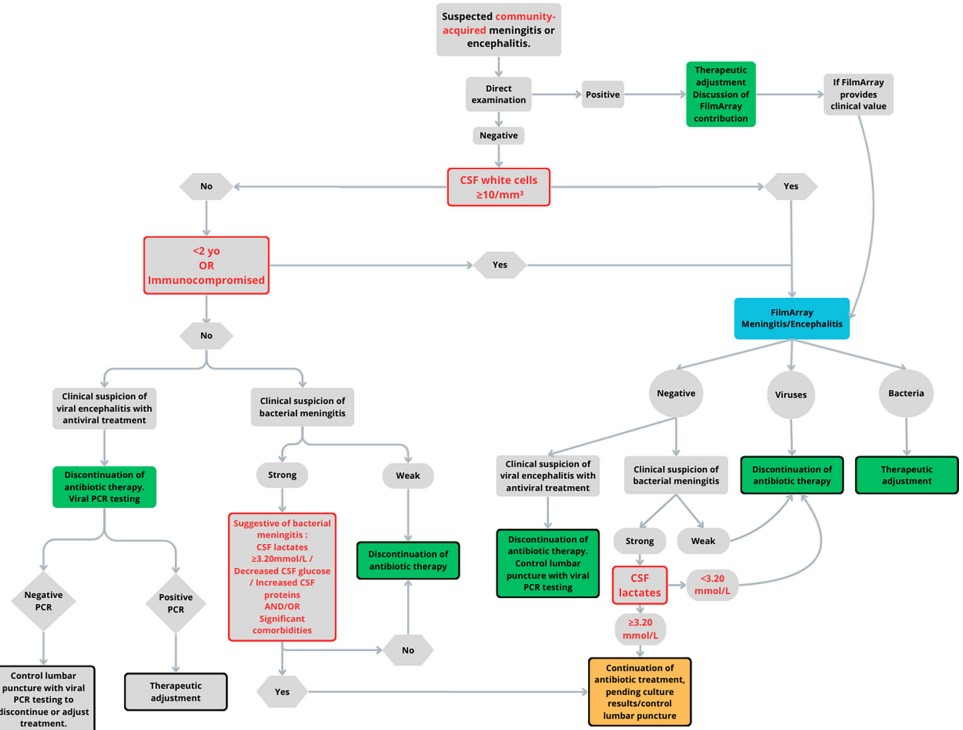

**FIG 4** Flowchart management of meningitis/encephalitis suspicion.

Applying this algorithm retrospectively would have avoided 261 (33%) FAME tests and could have increased early antibiotic discontinuation among treated patients with negative CSF and negative FAME from approximately 50% to over 64%, suggesting tangible stewardship and cost benefits. Only one Enterovirus case with a negative specific PCR—raising doubts about the clinical validity of this infection—would have been missed. Among the 12 cases in which the FAME panel yielded a negative result despite a positive CSF culture, only 7 were ultimately considered true meningitis, including 2 nosocomial cases. Lactate concentrations were available for only five of these patients. In one case classified as *Staphylococcus aureus* contamination, lactate levels were slightly above the threshold (3.8 mmol/L). In another case of presumed contamination, lactate was below the standard cut-off of 3.2 mmol/L, which would have supported antibiotic discontinuation. In three cases deemed to represent meningitis, lactate values were markedly elevated, justifying the continuation of antibiotic therapy despite a negative FAME result.

Limitations of our study include its retrospective design, heterogeneity of inclusion across centers, incomplete lactate measurements in some patients, and lack of independent adjudication of therapeutic decisions. A prospective evaluation of the proposed flowchart is required to validate safety, stewardship impact, and cost-effectiveness.

## Conclusion

We provide evidence-based, operational criteria for targeted FAME use and show that integrating CSF WBC and lactate with clinical context allows testing of patients most likely to benefit while enabling safe early discontinuation of unnecessary antimicrobial therapy. Systematic use of this algorithm in community-acquired ME, routine panel testing for immunocompromised patients and infants under 2 years, and reliance on WBC ≥10/mm$^3$ and lactate thresholds to guide testing and antibiotic decisions should improve antimicrobial stewardship, reduce costs, and preserve diagnostic sensitivity for clinically significant bacterial infections. Prospective validation in an independent cohort is warranted.

## ACKNOWLEDGMENTS

The authors thank all heads of departments for sharing their patients' data: Toulouse University Hospital: Prof. Isabelle Claudet, Prof. Sandrine Charpentier; Dr. Lionel Berthomieux, Dr. Sonia Pelluaux, Dr. Béatrice Riu-Poulenc, Dr. Erick Grouteau, Prof. Laurent Sailler, Dr. Thierry Seguin, Prof. Jean-Christophe Sol, Dr. Jean-François Albucher, Prof. Géraldine Gascoin, and Prof. Pierre Delobel. French Armed Forces Teaching Hospital Bégin: Dr. Andriamanantena Dinaherisoa, Dr. Dubost Clément, Dr. Woloch Alexandre, and Dr. Vanquaethem Hélène. Reims University Hospital: Prof. Firouze BaniSadr, Dr. Philippe Berger, Prof. François Boyer, Prof. Gaëtan Deslée, Dr. Béatrice Digeon, Prof. Stéphane Gennai, Dr. Juliette Jegou, Dr. Vincent Legros, Prof. Claude-Fabien Litré, Dr. Béatrice Monlibert, Prof. Solène Moulin, Prof. Christine Pietrement, Dr. Frédéric Pingot, Prof. Amélie Servettaz, Prof. Manuelle Viguier, and Dr. Pierre Willem. Lyon University Hospital: Dr. Marine Butin, Prof. Etienne Javouhey, Dr. Yves Gillet, and all heads of emergency and neurology departments. Saint Joseph Hospital: Dr. Fanny Autret, Dr. Philippe Azria, Dr. Cédric Bruel, Dr. Olivier Ganansia, and Dr. Mathieu Zuber. Poitiers Hospital: all heads of departments for which FAME were performed.

The GMC study group: Dr. Luc Deroche and Sébastien Lhomme.

The study was designed by N.N.-G., M.M., and T.G.. Patients were included by N.N.-G., M.M., C.P.-J., G.P.D.P., V.C., and A.-G.R. Data curation was performed by N.N.-G. and M.M. Formal analysis werewas performed by N.N.-G., M.M., A.M., and T.G. Writing—original draft was performed by N.N.-G., M.M., and T.G. Editing of the manuscript was performed by N.N.-G., M.M., A.M., and T.G. All authors contributed to final manuscript writing and approved the submitted version of the manuscript.

The FAMOuS study was registered as a MR004 study (MR00426052023) allowing manuscripts containing any individual person's data to be published.

## AUTHOR AFFILIATIONS

[1]INSERM, CHU de Reims, Laboratoire de Bactériologie-Virologie-Hygiène hospitalière, P3Cell, U 1250, Université de Reims Champagne-Ardenne, Reims, France

[2]CHU de Reims, Service de Médecine Interne, Maladies Infectieuses et Immunologie Clinique, Université de Reims Champagne-Ardenne, Reims, France

[3]Laboratoire de Bactériologie et d'Hygiène Hospitalière, CHU de Poitiers, Poitiers, France

[4]Service de Microbiologie Clinique, Hôpitaux Paris Saint-Joseph and Marie-Lannelongue, Paris, France

[5]Institut Micalis UMR 1319, Université Paris-Saclay, INRAe, AgroParisTech, Orsay, France

[6]Département de Biologie Médicale, Hôpital National d'Instruction des Armées (HNIA) BEGIN, Saint-Mandé, France

[7]Département de Bactériologie, Institut des Agents infectieux, Hospices Civils de Lyon, Lyon, France

[8]Service de Bactériologie-Hygiène Hospitalière, CHU de Toulouse, Hôpital Purpan, Institut de Recherche en Santé Digestive (IRSD), Université de Toulouse, INSERM, INRAE, ENVT, UPS, Toulouse, France

[9]Unité de Médecine Intensive et Réanimation, Université de Reims Champagne-Ardenne, CHU de Reims, Reims, France

## AUTHOR ORCIDs

Thomas Guillard  http://orcid.org/0000-0002-3795-0398

## AUTHOR CONTRIBUTIONS

Nathan Nicolau-Guillaumet, Conceptualization, Data curation, Formal analysis | Marin Moutel, Conceptualization, Data curation, Formal analysis | Anaëlle Muggeo, Formal analysis | Thomas Guillard, Conceptualization, Formal analysis, Writing – review and editing.

## ETHICS APPROVAL

Each patient received information about this study and was free to refuse to participate. The FAMOuS study was reviewed and approved by the Reims GDPR (General Data Protection Regulation) controller and registered as a MR004 study (MR00426052023), in agreement with the National Commission for Information Technology and Civil Liberties (CNIL MR004 conformity number: 2206749 v0).

## ADDITIONAL FILES

The following material is available online.

### Supplemental Material

**Supplemental material (Spectrum00566-26-s0001.docx).** Fig. S1–S4; Tables S1–S4.

### Open Peer Review

**PEER REVIEW HISTORY (review-history.pdf).** An accounting of the reviewer comments and feedback.

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
