## [Reviewer comments · Microbiology Spectrum]

Microbiology Spectrum

Optimized Use of the FilmArray® Meningitis/Encephalitis Panel for Early Discontinuation of Antibiotic Therapy

Nathan Nicolau-Guillaumet, Marin Moutel, Chloé Plouzeau, Gauthier Péan de Ponfilly, Virginie Courbin, Anne-Gaëlle RANC, Hélène REVILLET, bruno mourvillier, Maxime HENTZIEN, Anaëlle Muggeo, and Thomas Guillard

Corresponding Author(s): Thomas Guillard, Université de Reims Champagne-Ardenne

Review Timeline:

Submission Date:	March 11, 2026
Editorial Decision:	March 26, 2026
Revision Received:	March 29, 2026
Accepted:	April 3, 2026

Editor: Cheryl Andam

Reviewer(s): The reviewers have opted to remain anonymous.

Transaction Report:

DOI: <https://doi.org/10.1128/spectrum.00566-26>

Re: Spectrum00566-26 (Optimized Use of the FilmArray® Meningitis/Encephalitis Panel for Early Discontinuation of Antibiotic Therapy)

Dear Prof. Thomas Guillard:

I am pleased to inform you that your manuscript has been editorially accepted for publication. However, there are a few additional questions in the submission form that need to be answered before the final decision. Once these are completed, please return your submission so that I can move your paper to acceptance.

Please return the manuscript within 60 days; if you cannot complete the modification within this time period, please contact me. If you do not wish to submit the manuscript and prefer to submit it to another journal, notify me immediately so that the manuscript may be formally withdrawn from consideration by Spectrum.

Revision Guidelines

Sincerely,
Cheryl Andam
Editor
Microbiology Spectrum

Re: Spectrum00566-26R1 (Optimized Use of the FilmArray® Meningitis/Encephalitis Panel for Early Discontinuation of Antibiotic Therapy)

Dear Prof. Thomas Guillard:

Your manuscript has been accepted, and I am forwarding it to the ASM production staff for publication. Your paper will first be checked to make sure all elements meet the technical requirements. ASM staff will contact you if anything needs to be revised before copyediting and production can begin. Otherwise, you will be notified when your proofs are ready to be viewed.

Sincerely,
Cheryl Andam
Editor
Microbiology Spectrum